# Structure of the human astrovirus capsid spike in complex with the neonatal Fc receptor

Adam Lentz[1], Sarah Lanning[2], Khurshid R. Iranpur[1], Lena Ricemeyer [3], Carlos F. Arias[4] & Rebecca M. DuBois [3] ✉

Human astroviruses (HAstVs) are a leading cause of viral gastroenteritis in children worldwide. Recently the neonatal Fc receptor (FcRn) was identified as a receptor for HAstV, however the molecular basis for the FcRn-HAstV interaction remained unclear. Here, we report the crystal structure of FcRn bound to the HAstV capsid spike domain at 3.4 angstroms resolution. We show that all classical HAstV spikes bind to FcRn and we identify three conserved HAstV spike residues that mediate binding to FcRn. Using competition binding assays, we show that the HAstV spike competes with IgG for binding to FcRn. Additionally, we demonstrate that the FcRn inhibitor, nipocalimab, and anti-HAstV neutralizing monoclonal antibodies block HAstV spike binding to FcRn, revealing their neutralization mechanisms and supporting their therapeutic potential. Overall, our findings illuminate a crucial interaction in the HAstV life cycle, which may help to inform the development of a HAstV vaccine and antibody therapies.

Human astroviruses (HAstVs) are an important cause of viral diarrhea in children worldwide[1–3]. Between 2–9% of non-bacterial diarrhea in children can be attributed to HAstVs[4,5]. Seropositivity rates for HAstVs are high, with 90% of the population having antibodies towards at least one HAstV by the age of 9[6–10]. In addition to the classical HAstVs (HAstV1-8), the divergent HAstVs (MLB and VA/HMO clades) are also found in human stool samples worldwide[11–15]. Despite their global prevalence, HAstVs remain relatively poorly characterized, impeding the development of HAstV vaccines and therapeutics.

HAstVs are non-enveloped icosahedral viruses with positive-sense single-stranded RNA genomes. Their genomes range from 6–8 kilobases and contain three open reading frames, ORF1a, ORF1b, and ORF2. ORF1a and ORF1b encode nonstructural proteins, and ORF2 encodes the capsid protein. The classical HAstV capsid protein precursor has a highly basic amino terminus that binds to the viral genome, a core domain that encapsulates and protects the viral genome, a spike domain that forms dimeric globular spikes on the surface of the

virion, and a highly acidic carboxy terminus that is removed intracellularly by caspases[16–18]. The viral genome is encapsulated by 180 copies of the HAstV capsid to form the immature HAstV particle, presenting 90 dimeric spikes[19]. Extracellular proteolysis results in the removal of spikes and cleavage of the capsid surface[20], forming the mature HAstV particle, presenting only 30 dimeric spikes. While the reason for this loss of 60 spikes remains unclear, it may promote receptor binding and/or genome uncoating. Previous studies suggest that the HAstV capsid spike is involved in receptor engagement, and a number of conserved sites were proposed as potential receptor binding sites[21–23]. In addition, neutralizing monoclonal antibodies (mAbs) targeting the HAstV spike can block virus attachment to human cells[10,23–25], further highlighting its potential involvement in receptor binding. However, without the identification of a receptor for HAstVs, these ideas have remained unconfirmed.

Recently, the neonatal Fc receptor (FcRn) was identified in two separate studies as a functional receptor for HAstVs[26,27]. These studies

[1]Department of Microbiology & Environmental Toxicology, University of California Santa Cruz, Santa Cruz, California, USA. [2]Department of Molecular Cell and Developmental Biology, University of California Santa Cruz, Santa Cruz, California, USA. [3]Department of Biomolecular Engineering, University of California Santa Cruz, Santa Cruz, California, USA. [4]Departamento de Genética del Desarrollo y Fisiología Molecular, Instituto de Biotecnología, Universidad Nacional Autónoma de México, Cuernavaca, Morelos, Mexico. ✉e-mail: rmdubois@ucsc.edu

demonstrate that deleting FcRn significantly reduces the propagation of all eight classical HAstVs in human colon carcinoma (Caco-2) cells, the model cell line for HAstV infection, as well as in intestinal organoids. In addition, in cells that are normally non-permissive to HAstV infection, FcRn expression can promote HAstV infection. These studies also show that recombinant FcRn can directly bind recombinant capsid spikes from HAstV1, HAstV4, and HAstV8. Collectively, these data clearly demonstrate that FcRn is a receptor for HAstV.

FcRn is an Fc gamma receptor that binds the Fc region of IgG antibodies. FcRn is a heterodimeric receptor formed by a heavy chain (FcRn α-chain, or αFcRn) and a light chain (β2 microglobulin, or β2M). FcRn was initially identified and named for its role in translocating maternal IgG across the neonate gut epithelium[28]. However, FcRn has since been recognized to function outside the gut and well beyond the neonatal stage of life. One key function of FcRn occurs in the endothelial cells lining blood vessels. Here, FcRn acts as a cellular recycling mechanism by salvaging IgG as well as human serum albumin (HSA) from intracellular degradation and releasing them back into circulation, dramatically increasing the serum half-life of both proteins[29,30]. This recycling of IgG and HSA is driven by pH-dependent binding. FcRn binds IgG and HSA optimally at the acidic pH of the endosome (pH 6.5-5.5) and redirects them back to the cell surface for release at neutral pH[31].

HAstVs are part of a growing list of diverse viruses that use FcRn as a receptor. FcRn was also identified as a receptor for echoviruses, which are nonenveloped viruses in the *Enterovirus B* group of the *Picornaviridae* family[32]. FcRn functions largely as the uncoating receptor for these viruses, although for at least echovirus 18, FcRn is both the attachment and uncoating receptor[33]. Remarkably, acidic pH and engagement with FcRn are the only requirements for uncoating these echoviruses[33]. In addition, FcRn is a receptor for multiple arteriviruses, including porcine reproductive and respiratory syndrome virus (PRRSV)[34,35]. Unlike the enteroviruses and HAstVs, arteriviruses are enveloped viruses, revealing that diverse viruses have evolved to utilize FcRn to promote infections. For PRRSV, FcRn appears dispensable for virus attachment and internalization, but may have a role in uncoating and genome release[35].

Here, we report the crystal structure of FcRn in complex with the HAstV1 capsid spike domain. This structure defines the molecular interactions governing HAstV binding to FcRn, revealing a previously unrecognized site of vulnerability on the surface of the HAstV virion. Binding affinity studies provide functional insights into the breadth and pH dependence for FcRn binding, and competition binding studies provide insights into the mechanism of antibody neutralization. Altogether, these studies increase our understanding of HAstV infection and provide a foundation for the development of a HAstV vaccine and antiviral therapies.

## Results

### FcRn binds HAstV1 capsid spike at neutral and acidic pH and forms a stable complex

FcRn binds IgG and HSA optimally at mildly acidic pH, but the effect of pH on FcRn-HAstV spike binding is unknown. To determine if mildly acidic pH is also optimal for the FcRn-HAstV1 spike interaction, we generated recombinant FcRn and recombinant HAstV1 spike and used biolayer interferometry (BLI) to determine the binding dissociation constants ($K_D$) at pH 7.0 and pH 5.0 (Fig. 1a). The $K_D$ at pH 7.0 is 188 nM, whereas at pH 5.0 it is 102 nM, indicating that the FcRn-HAstV1 spike interaction occurs at both neutral and mildly acidic pH but is slightly stronger at mildly acidic pH.

To understand the binding stoichiometry of the FcRn heterodimer to the HAstV spike homodimer, purified recombinant FcRn and HAstV1 spike were pre-incubated overnight and then loaded onto a size exclusion chromatography (SEC) column. The SEC trace revealed two peaks (Fig. 1b). When compared to gel filtration standards, the first

peak eluted at ~145 kDa. SDS-PAGE revealed bands for FcRn (αFcRn and β2M chains) and for HAstV1 spike (Fig. 1c), indicating that the first peak is the complex (Fig. 1b). The HAstV1 spike - FcRn complex size estimation of ~145 kD suggests that two FcRn heterodimers bind to one HAstV spike homodimer (a 2:2 ratio of FcRn:spike).

### The structure of FcRn in complex with HAstV1 spike reveals a poorly conserved binding site

To understand the molecular basis for HAstV spike binding to FcRn, we crystallized and solved the 3.4Å-resolution crystal structure of the HAstV1 spike - FcRn complex (Fig. 2a, Supplementary Table 1 and Supplementary Fig. 1). Consistent with the size and ratio estimated by SEC, the structure reveals that two FcRn heterodimers bind to one HAstV1 spike homodimer (Fig. 2a). There were two complexes within the crystallographic asymmetric unit, with the most dynamic regions being the membrane proximal regions of FcRn (Supplementary Fig. 2). At the ~570 Å² binding interface, two loops on the HAstV1 spike wrap around FcRn residue W154. The W154 side chain sits in a hydrophobic pocket formed by HAstV1 spike residues A469, V472, Y475, and L509 (Fig. 2a top view). Similarly, HAstV1 spike residues V472 and I512 sit in hydrophobic pockets on FcRn. Surrounding these binding pockets are several hydrogen bonds and an electrostatic interaction (Fig. 2a). The negatively charged FcRn residue E138 forms an electrostatic interaction with the positively charged HAstV1 spike residue K467, and it also forms a hydrogen bond with Y475. The FcRn residue N136 side chain forms a hydrogen bond with HAstV1 spike residue D473. The remaining hydrogen bonds are between FcRn residues D153, W154, and E156 sidechains and the backbone of HAstV1 spike residues K514, N511, and V472, respectively. Finally, the HAstV1 spike - FcRn binding interaction is capped by β2M residue I21 contacting HAstV1 spike residues V472, D473, and Y475, forming ~40 Å² of the interface.

Remarkably, the binding interface on the HAstV spike domain is poorly conserved (Fig. 2b). The sequence alignment of the binding interface for the eight classical HAstV spikes reveals that there are only two strictly conserved amino acid residues across the eight classical HAstV spikes at this interface, Y475 and K514. Residue K467 is another notable residue with only 75% conservation across the eight serotypes, but 100% conservation as a positively charged residue. These three residues may form key interactions that enable sequence-diverse HAstV spikes to bind FcRn.

### FcRn binds the eight classical HAstV spikes with varying affinity

Given that FcRn binds a poorly conserved region of the HAstV capsid spike domain, we sought to determine the $K_D$ for binding to FcRn for each of the eight classical HAstV spikes and two divergent spikes from HAstV-VA1 and HAstV-MLB1 (Table 1 and Supplementary Fig. 3). All recombinant HAstV spikes were purified by SEC and, compared to gel filtration standards, have been shown to form dimers[36-38]. FcRn binds to HAstV1, 4, 5, 6, and 8 spikes with similar $K_D$s in the hundred-nanomolar range. Interestingly, FcRn affinities to HAstV2, 7, and 3 spikes are weaker, with $K_D$s in the micromolar, ten-micromolar, and hundred-micromolar ranges, respectively. Notably, we did not observe any measurable binding signal (tested up to 2 µM FcRn) to the divergent HAstV VA1 or MLB1 spikes, which is not surprising given their absence of sequence homology with classical HAstV spikes and strikingly different structures[36,37].

### HAstV1 spike K467A, Y475A, and K514A mutants have reduced binding to FcRn

We next investigated the importance of the three most conserved HAstV spike residues at the FcRn-HAstV spike interface (K467, Y475, and K514) for binding to FcRn. We generated recombinant HAstV1 spike mutants K467A, Y475A, and K514A, analyzed the mutants by SEC to confirm dimer formation (Supplementary Fig. 4) suggesting proper display of the FcRn binding site, and determined the $K_D$s for spike mutants binding

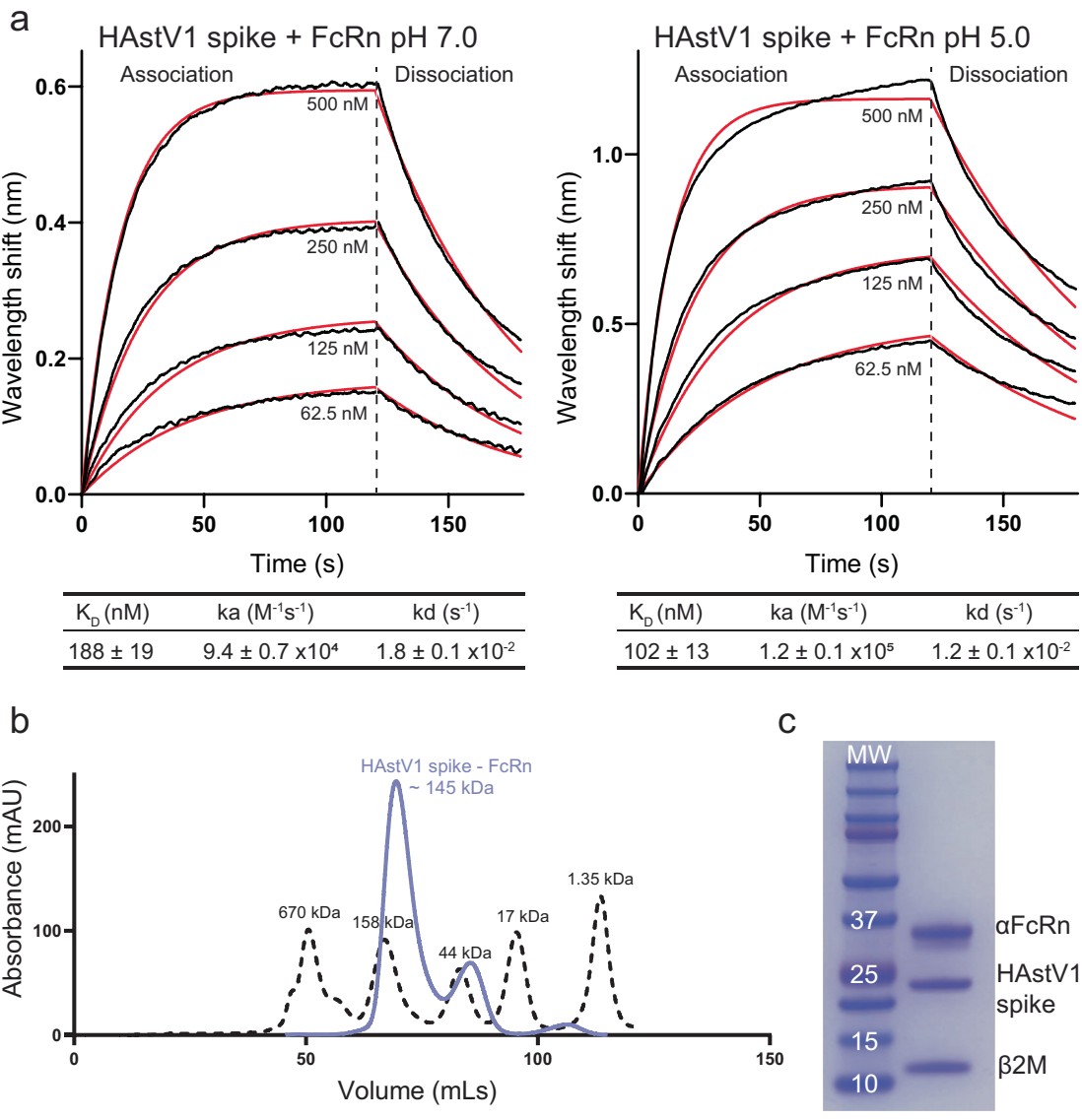

Fig. 1 | FcRn binds HAstV1 spike at pH 7.0 and pH 5.0 and forms a complex in solution. a Representative BLI traces of streptavidin biosensors loaded with biotinylated HAstV1 spike at pH 7.0 (left) or pH 5.0 (right) and dipped into a 2-fold dilution series of FcRn ranging from 500-62.5 nM. Binding signals are shown as black lines, and curve fits are shown as red lines. Binding equilibrium constant ($K_D$), association rate (ka), and dissociation rate (kd) are reported as an average of three replicates ± standard deviation. b SEC trace (solid purple line) overlayed with BioRad gel filtration standards (dashed black line). c Coomassie-stained SDS-PAGE gel image of SEC fraction from FcRn·HAstV1 spike complex peak. The image is representative of two independent SDS-PAGE experiments on this sample. Source data are provided as a source data file.

to FcRn (Table 2 and Supplementary Fig. 5). We also generated a triple mutant with all three residues mutated to alanine, referred to as 3 A (Table 2 and Supplementary Figs. 4 and 5). The K467A mutant was predicted to affect binding, given that the spike would lose the electrostatic interaction formed between K467 and FcRn residue E138. The Y475A mutant was also expected to affect binding given its interactions with FcRn residues W154 and E138. It was unclear how K514A would affect binding since the main interaction with FcRn is with its backbone, although the K514 side chain forms an electrostatic interaction with nearby HAstV1 spike residue D517, which may indirectly support the β7-β8 loop containing FcRn-interacting residues. As predicted, the K467A mutant showed dramatically reduced binding to FcRn with a $K_D$ of 15.0 μM. Mutant Y475A also had notably reduced binding to FcRn with a $K_D$ of 8 μM. Mutant K514A binding to FcRn was reduced ~10-fold with a $K_D$ of 2.5 μM. There was no detectable binding signal for the triple mutant 3 A (tested up to 2 μM FcRn), revealing the importance of these three conserved residues for binding to FcRn.

## HAstV1 spike competes with IgG, but not HSA, for binding to FcRn

FcRn engages IgG and HSA in endosomes at acidic pH and recycles them by bringing them back to the cell surface for their release at neutral pH. FcRn can bind to both IgG and HSA at the same time in vitro, due to distinct binding sites on FcRn[39,40] (Fig. 3a). It is reasonable to speculate that HAstV may need to compete with IgG and/or HSA in an infection context to engage FcRn. We therefore tested if HAstV1 spike can bind FcRn in the presence of IgG or HSA at pH 5.0. Our crystal structure suggests that HAstV1 spike may need to compete for binding to FcRn with IgG but not HSA, since HAstV1 spike and IgG bind the same region of FcRn (Fig. 3a). Using BLI competition binding assays, we evaluated IgG and HSA binding to FcRn that was pre-complexed with HAstV1 spike, or vice versa (Fig. 3c and Supplementary Fig. 6). Consistent with our crystal structure, HSA is still able to bind FcRn pre-complexed with HAstV1 spike, reaching 77% of the control binding signal (HSA binding to FcRn in the absence of HAstV1 spike).

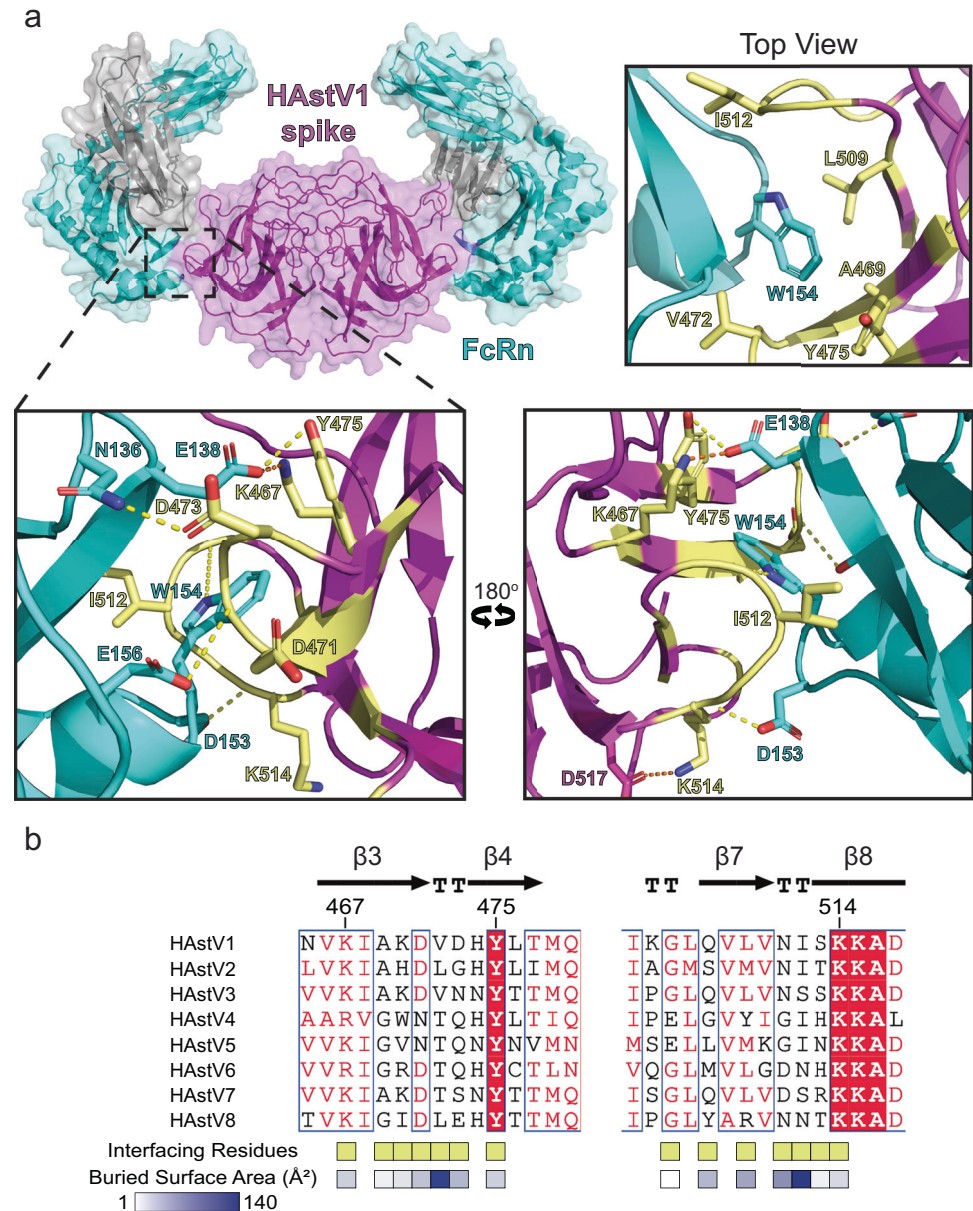

**Fig. 2 | Crystal structure of FcRn in complex with HAstV1 spike. a** Cartoon ribbon structure of the FcRn-HAstV1 spike complex with surface shown at 50% transparency. HAstV1 spike is shown in magenta and FcRn shown in teal (αFcRn) and gray (β2M). The binding interface is shown from the top and both sides of the complex. Interfacing residues on the HAstV1 spike are colored pale yellow, and notable residues on FcRn and the HAstV1 spike are shown as sticks and labeled. Hydrogen bonds are shown as dashed yellow lines, and electrostatic interactions are shown as orange dashed lines. FcRn residue numbering is based on Uniprot accession number P55899. **b** Sequence alignment of FcRn binding site regions of the eight classical HAstV spikes made with the Clustal Omega and ESPript servers. Interfacing residues are denoted with pale yellow boxes below the alignment. Buried surface area for each interfacing residue was determined with the PDBePISA server and shown with a box below the alignment, colored with a gradient from white (less buried) to blue (more buried). The three most conserved interfacing residues are labeled with the HAstV1 capsid protein sequence number (Uniprot accession number O12792) above the alignment. Secondary structure from HAstV1 spike is shown above the alignment. Source data are provided as a source data file.

Similarly, HAstV1 spike is still able to bind to FcRn pre-complexed with HSA, reaching 84% of the control binding signal. In contrast, IgG cannot bind to FcRn pre-complexed with HAstV1 spike, only reaching 19% of the control binding signal. Similarly, HAstV1 spike cannot bind to FcRn pre-complexed with IgG, reaching 3% of the control binding signal.

### Nipocalimab blocks HAstV1 spike binding to FcRn

Nipocalimab is an FDA-approved human IgG1 monoclonal antibody that binds to the IgG binding site on FcRn at both neutral and acidic pH and blocks IgG binding[41]. Nipocalimab's binding site on FcRn also overlaps with the HAstV1 spike binding site (Fig. 3b), suggesting that nipocalimab may block HAstV1 spike binding to FcRn. Using BLI competition binding assays, we evaluated HAstV1 spike, IgG, and HSA binding to FcRn that was pre-complexed with nipocalimab (Fig. 3c and Supplementary Fig. 6c). HSA is still able to bind FcRn pre-complexed with nipocalimab, reaching 99% of the control binding signal. However, consistent with structural evidence, HAstV1 spike cannot bind to FcRn pre-complexed with nipocalimab, only reaching 26% of the control binding signal. To confirm and extend these findings, we

**Table 1 | Binding affinities of HAstV spikes to FcRn at pH 7.0**

| Genotype | $K_D$ (µM) | ka (M⁻¹s⁻¹) | kd (s⁻¹) | X² | R² |
|---|---|---|---|---|---|
| HAstV1 | 0.27 ± 0.02 | $5.2 ± 0.2 × 10^4$ | $1.4 ± 0.1 × 10^{-2}$ | 0.296 | 0.997 |
| HAstV2 | 1.7 ± 0.5 | $3.1 ± 0.9 × 10^4$ | $4.8 ± 0.3 × 10^{-2}$ | 0.126 | 0.996 |
| *HAstV3 | 181 ± 85 | $6.5 ± 2.3 × 10^2$ | $9.9 ± 0.2 × 10^{-2}$ | 0.144 | 0.772 |
| HAstV4 | 0.23 ± 0.02 | $4.7 ± 0.3 × 10^4$ | $1.1 ± 0.1 × 10^{-2}$ | 0.327 | 0.997 |
| HAstV5 | 0.152 ± 0.004 | $3.9 ± 0.1 × 10^4$ | $6.0 ± 0.1 × 10^{-3}$ | 0.633 | 0.996 |
| HAstV6 | 0.5 ± 0.1 | $3.1 ± 0.3 × 10^4$ | $1.5 ± 0.1 × 10^{-2}$ | 0.345 | 0.990 |
| *HAstV7 | 10.3 ± 3.2 | $1.4 ± 0.4 × 10^3$ | $1.3 ± 0.1 × 10^{-2}$ | 0.023 | 0.986 |
| HAstV8 | 0.39 ± 0.03 | $7.6 ± 0.3 × 10^4$ | $3.0 ± 0.2 × 10^{-2}$ | 0.145 | 0.997 |
| VA1 | NB | NB | NB | NB | NB |
| MLB1 | NB | NB | NB | NB | NB |

*Kinetics estimated from one FcRn concentration (2 µM).
NB = no detectable binding at the tested concentrations.

**Table 2 | Binding affinities of mutant HAstV spikes to FcRn at pH 7.0**

| Mutant | $K_D$ (µM) | ka (M⁻¹s⁻¹) | kd (s⁻¹) | X² | R² |
|---|---|---|---|---|---|
| WT | 0.27 ± 0.02 | $5.2 ± 0.2 × 10^4$ | $1.4 ± 0.1 × 10^{-2}$ | 0.296 | 0.997 |
| *K467A | 15.0 ± 0.7 | $1.9 ± 0.1 × 10^4$ | $2.84 ± 0.02 × 10^{-1}$ | 0.051 | 0.983 |
| *Y475A | 8 ± 4 | $3.7 ± 1.9 × 10^4$ | $2.2 ± 0.4 × 10^{-1}$ | 0.099 | 0.971 |
| K514A | 2.5 ± 0.3 | $2.4 ± 0.3 × 10^4$ | $5.6 ± 0.3 × 10^{-2}$ | 0.088 | 0.996 |
| 3 A | NB | NB | NB | NB | NB |

*Kinetics estimated from one FcRn concentration (2 µM).
NB = no detectable binding at the tested concentrations.

developed a competitive ELISA to evaluate nipocalimab blocking of HAstV1 spike binding to FcRn (Fig. 3d). Varying concentrations of nipocalimab were tested against 125 nM HAstV1 spike. Nipocalimab at the equivalent concentration of 125 nM completely blocks HAstV1 spike binding to FcRn, and even low picomolar concentrations resulted in a significant reduction in HAstV1 spike binding (Fig. 3d).

### Neutralizing antibodies block FcRn binding to HAstV spike

A panel of five neutralizing monoclonal antibodies (mAbs) has been described previously[42]. Using structural studies, these mAbs were found to target several non-overlapping epitopes on the HAstV spike[10,25]. Despite the distinct epitopes, all five mAbs block HAstV attachment to Caco-2 cells, and three of those mAbs were recently shown to block FcRn binding to HAstV spike[10,25]. We tested whether the other two mAbs, 2D9 and 3E8, can block FcRn binding to HAstV8 spike using BLI competition binding assays. Instead of testing full-length mAbs, which contain an Fc domain with a binding site for FcRn, we generated single-chain variable fragments (scFvs) to assess antibody competition with FcRn for binding to HAstV8 spike. HAstV8 spike was pre-complexed with scFv 2D9 or 3E8 and tested for binding to FcRn. Both scFv 2D9 and 3E8 prevented FcRn binding to HAstV8 spike, reaching only ~1–15% of the control binding signal (FcRn binding to HAstV8 spike in the absence of scFv 2D9 or 3E8) (Fig. 4a and Supplementary Fig. 5). These results support that blocking FcRn binding may be the mode of neutralization for these mAbs.

### HAstV spike can bind mouse FcRn

The HAstV field has historically been hindered by not having an animal model for HAstV infection. HAstV does not infect mice, and one explanation might be that the HAstV spike cannot bind mouse FcRn. To test this, we used BLI binding assays to evaluate HAstV1 spike binding to recombinant mouse FcRn (Supplementary Fig. 8).

Surprisingly, we find that HAstV1 spike binds to mouse FcRn with a slightly stronger affinity than to human FcRn. This difference appears to be driven by a slower off-rate. The interfacing region of FcRn is nearly 100% conserved between human and mouse FcRn. One potentially important change is mouse FcRn has glutamic acid at position 153, where human FcRn has aspartic acid. This longer side chain may allow it to reach further and form a stronger interaction with the HAstV1 spike. Another possible explanation for the slightly higher affinity towards mouse FcRn is an N-linked glycosylation site (N149 and T151) in mouse FcRn that is not present in human FcRn (T149 and G151). To test the impact that these two differences have on binding to HAstV1 spike, we generated three recombinant human FcRn mutants to present the mouse FcRn binding interface on human FcRn; D153E, T149N_G151T, and T149N_G151T_D153E. We determined the $K_D$s for FcRn mutants binding to HAstV1 spike at pH 7.0 (Supplementary Table 2 and Supplementary Fig. 9). Mutants D153E and T149N_G151T had ~10-fold and ~2-fold reduction in affinity with $K_D$s of 1.6 µM and 0.5 µM, respectively, compared to wild-type FcRn's $K_D$ of 0.23 µM. In addition, the triple mutant (T149N_G151T_D153E) bound even weaker to HAstV1 spike, suggesting that the slightly stronger binding affinity of HAstV1 spike for mouse FcRn may be due to a more global structural difference. Unless the slightly stronger interaction with mouse FcRn is detrimental towards infection, there is likely another reason why HAstV does not infect mice.

## Discussion

Here, we reported the crystal structure of FcRn in complex with the HAstV1 capsid spike at 3.4 Å resolution. Previous studies identified several sites of conservation on the HAstV spike dimer that could be potential receptor binding sites[21,23]. While these sites may be important for interaction with another host factor, FcRn does not bind at either site but instead binds to a largely non-conserved region on the side of the spike dimer (Fig. 2). This position allows for two FcRn molecules to bind to one HAstV spike homodimer. Notably, this 2:2 (FcRn:HAstV spike) binding stoichiometry mimics FcRn's interaction with IgG, which similarly has two binding sites for FcRn. While it is unclear how many FcRn receptors engage the HAstV virion in vivo, microscopic studies investigating FcRn localization reveal clustering of FcRn on cell membranes[43,44], suggesting that 2:2 interactions might occur.

We demonstrate that all eight classical HAstV spikes bind to FcRn. The divergent HAstV-MLB1 and HAstV-VA1 spikes do not bind FcRn, which is consistent with their strikingly different sequences and structures[36,37] and suggests that the divergent HAstVs may use different host factors to promote virus entry. Given the low sequence conservation in classical HAstVs at the FcRn binding site, it is not surprising that FcRn binds to the different classical HAstV spikes with

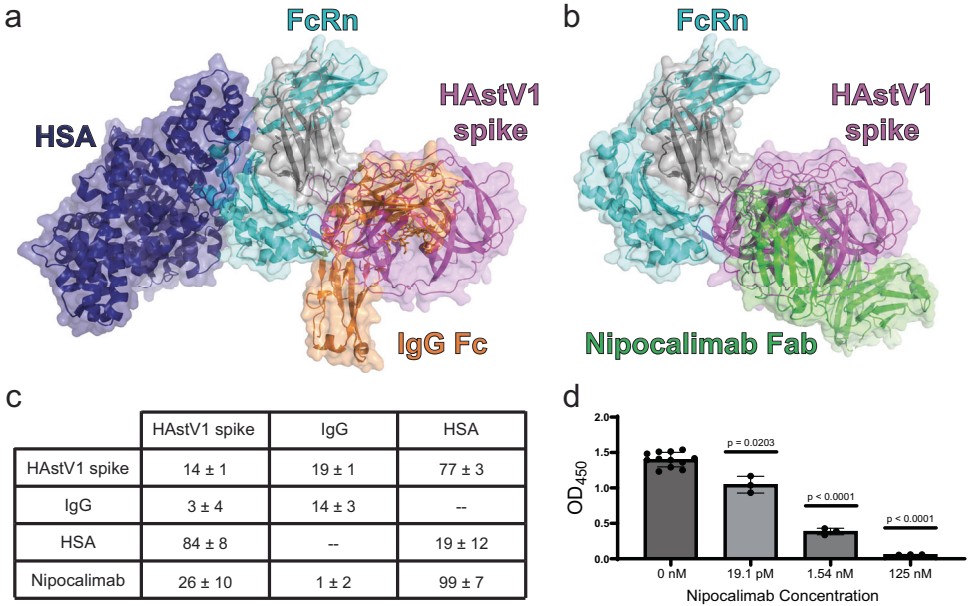

| | HAstV1 spike | IgG | HSA |
|---|---|---|---|
| HAstV1 spike | 14 ± 1 | 19 ± 1 | 77 ± 3 |
| IgG | 3 ± 4 | 14 ± 3 | -- |
| HSA | 84 ± 8 | -- | 19 ± 12 |
| Nipocalimab | 26 ± 10 | 1 ± 2 | 99 ± 7 |

**Fig. 3 | HAstV1 spike competes with IgG for binding to FcRn and nipocalimab blocks HAstV1 spike binding to FcRn. a** The structure of FcRn in complex with HAstV1 spike overlayed with the structure of FcRn in complex with IgG Fc and HSA (PDB: 4N0U). HAstV1 spike is shown in magenta, FcRn is shown in teal (αFcRn) and gray (β2M), HSA is shown in dark blue, and IgG Fc is shown in orange. **b** The structure of FcRn in complex with HAstV1 spike overlayed with the structure of FcRn in complex with the nipocalimab antigen binding fragment (Fab) (PDB: 9MI6). HAstV1 spike is shown in magenta, FcRn is shown in teal (αFcRn) and gray (β2M), and nipocalimab Fab is shown in green. **c** Binding competition table. Column 1 shows which protein was pre-complexed with FcRn, and row 1 shows which

protein was assessed for binding to the respective complex at pH 5.0. The values reported are percentages of the corresponding control binding signal from an FcRn-loaded biosensor with no pre-complexing. Values reported are an average of three replicates ± standard deviation. **d** Competitive ELISA results from mixing biotinylated HAstV1 spike at 125 nM with varying concentrations of nipocalimab and adding these pre-mixtures to an FcRn-coated ELISA plate at pH 7.0. After washing, HRP-streptavidin was used to detect bound HAstV1 spike. Results are an average of three replicates, and error bars represent standard deviation. Each nipocalimab concentration was compared to the 0 nM condition using a two-sided Welch's $t$ test. Source data are provided as a source data file.

varying affinities (Table 1). FcRn has the highest affinity for the spike from HAstV5, followed by HAstV4, HAstV1, HAstV8, HAstV6, HAstV2, HAstV7, and lastly HAstV3. One potential contributor to weaker binding affinity by HAstV7 and HAstV3 spikes is residue 512. In HAstV1, this residue is isoleucine, a hydrophobic residue nestled in a hydrophobic pocket on FcRn. HAstV3 and HAstV7 have a serine in this position, which introduces a small polar group that may be less favorable. However, in general, it is difficult to pinpoint specific residues that may be responsible for strong or weak affinity for FcRn, and extensive mutagenesis studies may be required to explain these binding affinity differences. Instead, in this study, we chose to investigate the three highly conserved HAstV residues by mutating them to alanine and assessing binding to FcRn. We found that single-point mutations in HAstV spike resulted in 10–50-fold weaker affinities for FcRn compared to wild-type HAstV1 spike (Table 2). Moreover, the triple alanine mutant 3 A binding affinity could not be determined at the concentrations tested, suggesting that collectively these three residues are key drivers of FcRn binding.

FcRn binds to IgG under acidic conditions. This is in part due to multiple histidine residues on IgG that interact with negatively charged residues on FcRn. One critical interaction occurs between FcRn residue E138 and IgG residue H310[40]. These two residues form an electrostatic interaction that is a main driver of the FcRn-IgG interaction. For this interaction to occur, H310 must be protonated, which occurs around pH 6.0 and lower. In the HAstV spike interaction with FcRn, E138 is also involved in forming an important electrostatic interaction with HAstV1 residue K467. However, unlike the IgG residue H310, HAstV residue K467 carries a positive charge at both neutral and acidic pH, possibly explaining the ability of HAstV1 spike to bind FcRn at pH 7.0 and pH 5.0. To test this idea, we generated recombinant HAstV1 spike with K467 mutated to histidine, and determined it's $K_D$ for binding FcRn at

pH 7.0 and pH 5.0 (Supplementary Fig. 10). As expected, the HAstV1 spike mutant K467H showed reduced binding affinity for FcRn at pH 7.0 with a $K_D$ of 22.5 μM, while at pH 5.0 the $K_D$ improved to 0.8 μM. While the $K_D$ did not return to wild-type levels at pH 5.0, possibly due to histidine being a shorter and less flexible side chain than lysine, the K467H mutant demonstrated pH-influenced binding. This reveals the potential competitive advantage that HAstV has towards binding FcRn at neutral pH. In an infection context, this advantage may be most pronounced at the more neutral pH of the gut epithelial surface. However, under the more acidic conditions of an endosome, HAstV may still have to compete with IgG, making competition with intestinal IgG a potential early barrier for HAstV infection. This study demonstrates that IgG and HAstV1 spike compete for binding to FcRn at acidic pH. This is not entirely surprising given that IgG and HAstV1 spike bind to the same region of FcRn, and the $K_D$ for FcRn binding to IgG1 is 760 nM[45], which is on a similar order of magnitude as the $K_D$ for HAstV1 spike. However, HAstV may have another binding advantage due to avidity via binding of multiple spikes to cellular FcRn. This may explain why a previous study showed that the presence of IgG during infection had no impact on HAstV propagation in Caco-2 cells[26].

A previous study demonstrated that the FDA-approved FcRn inhibitor nipocalimab prevents HAstV infection of Caco-2 cells, FcRn-expressing HEK293T cells, and human intestinal enteroids[27]. In this study, we provide mechanistic insight with nipocalimab directly blocking HAstV1 spike binding to FcRn by competitive BLI and ELISA, consistent with our structural evidence (Fig. 3c, d and Supplementary Fig. 6). Together, these findings further support nipocalimab and other FcRn inhibitors as potential therapeutics against classical HAstV infections. Future FcRn mutagenesis studies could further support the importance of this site in HAstV infection.

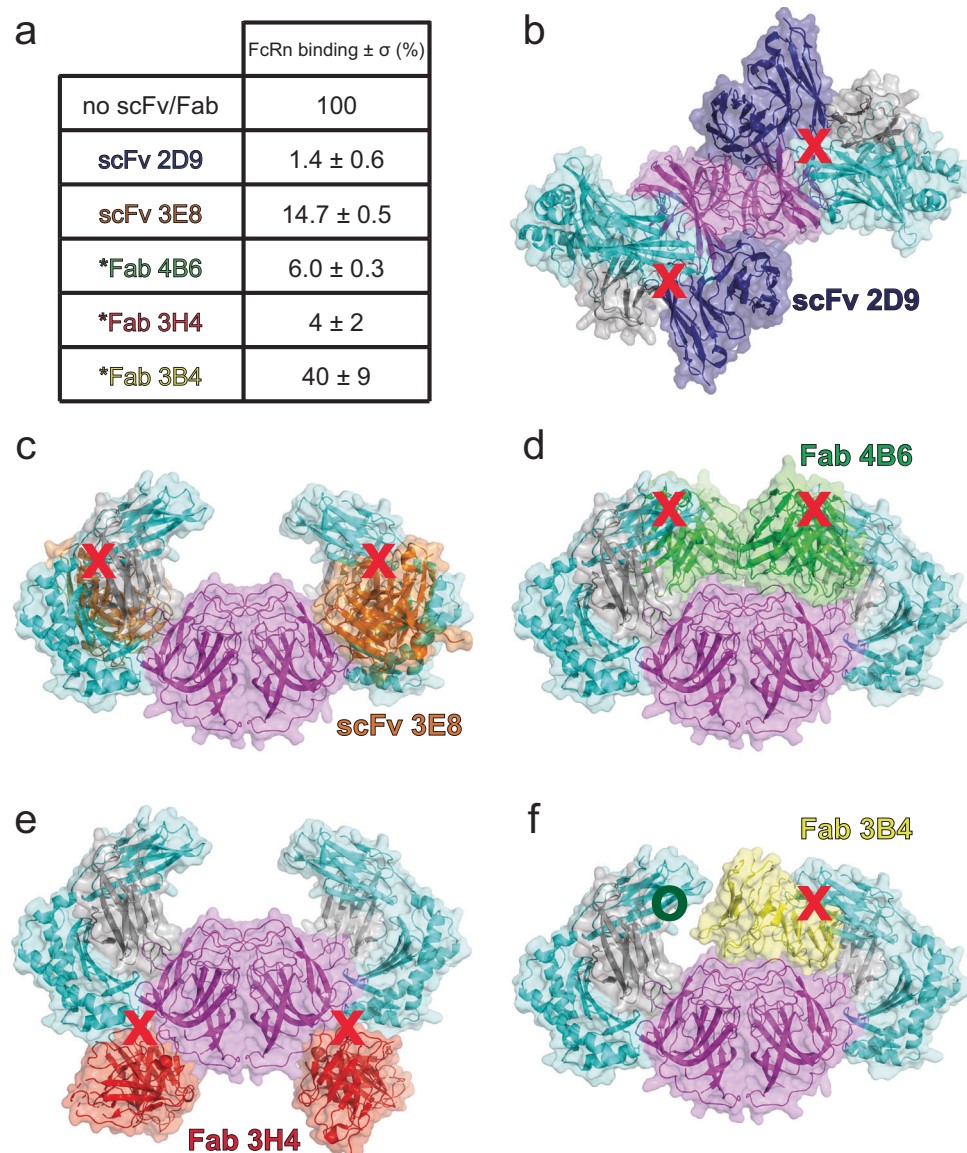

**Fig. 4 | Antibodies 2D9 and 3E8 block FcRn binding to HAstV8 spike. a** Binding competition table. After loading biosensors with HAstV8 spike, column one shows which antibody scFv was pre-complexed before assessing FcRn binding at pH 7.0. The values reported are percentages of a corresponding control binding signal from an HAstV8 spike-loaded biosensor with no pre-complexing. Values reported are an average of three replicates ± standard deviation. Values with an asterisk (*) were from binding competition studies reported previously[25]. **b** Top view of the structure of FcRn bound to HAstV1 spike overlayed with the structure of scFv 2D9 bound to HAstV8 spike (PDB: 7RK2). **c** Side view of the structure of FcRn in complex with HAstV1 spike overlayed with the structure of scFv 3E8 bound to HAstV8 spike (PDB: 7RK1). **d** Side view of the structure of FcRn in complex with HAstV1 spike overlayed with the structure of Fab 4B6 bound to HAstV2 spike (PDB: 9CN2). **e** Side view of the structure of FcRn in complex with HAstV1 spike overlayed with the structure of Fab 3H4 bound to HAstV1 spike (PDB: 9CBN). **f** Side view of the structure of FcRn in complex with the structure of FcRn in complex with HAstV1 spike overlayed with the structure of Fab 3B4 bound to HAstV1 spike (PDB: 9CBN). HAstV1 spike in magenta, FcRn in teal (αFcRn) and gray (β2M), scFv 2D9 in dark blue, scFv 3E8 in orange, Fab 3B4 in yellow, Fab 4B6 in green, and Fab 3H4 in red. Antibody clashes with FcRn marked with a red "x", and the absence of clashes marked with green "o". In structure images with Fabs, only the variable region is displayed. Source data are provided as a source data file.

Neutralizing mAbs target the HAstV spike domain and block HAstV infection in Caco-2 cells[10], but the molecular basis for neutralization remained unclear. Here we show that scFv 2D9 and scFv 3E8 block FcRn binding to HAstV8 spike (Fig. 4a). The 2D9 epitope just barely overlaps with the FcRn binding site (Fig. 4b), but it is sufficient to block FcRn binding. In contrast, the 3E8 epitope overlaps significantly with the FcRn binding site, explaining its ability to block FcRn binding (Fig. 4c). In addition, our structure allows interpretation of previous binding competition studies with three other antibodies (Fig. 4a)[25]. While the 4B6 epitope does not overlap with the FcRn binding site, Fab 4B6 sterically clashes with the membrane proximal region (α-3 domain) of αFcRn, explaining its ability to block FcRn binding (Fig. 4d). Similar to 2D9, the 3H4 epitope just barely overlaps with the FcRn binding site (Fig. 4e), but it is sufficient to block FcRn binding. Lastly, the 3B4 epitope does not overlap with the FcRn binding site, but Fab 3B4 sterically clashes with the membrane proximal region (α-3 domain) of αFcRn. However, since 3B4 binds in an asymmetric manner at the top of the spike, it only sterically clashes with one of the two FcRn molecules (Fig. 4f), explaining why Fab 3B4 only partially blocks FcRn binding (Fig. 4a). Altogether, these studies reveal how neutralizing mAbs sterically block HAstV spike binding to FcRn.

In one of the reports that identified FcRn as a receptor for HAstV, the authors noted that immature HAstV provides a stronger signal than mature HAstV for binding to FcRn in an ELISA[26]. This intriguing result led us to align our FcRn-HAstV1 spike structure with the spikes on the cryoelectron microscopy models of the immature and mature HAstV virion (Supplementary Fig. 11)[19,22]. With 30 total spikes, the mature virus has 60 possible binding sites for FcRn. Our modeling suggests that all sites are accessible, and multiple FcRn molecules could bind to multiple spikes at the same time. On the other hand, the immature virus has 90 total spikes and 180 possible binding sites for FcRn. For the 30 spikes that remain on the mature virus, which reside on the two-fold symmetry axes, FcRn binding to these spikes would be blocked by the spikes on the five-fold and three-fold axes. This leaves 60 spikes for FcRn to bind to on the immature virus, and all remaining 120 binding sites appear to be accessible, which may explain the reported ELISA data[26]. In a larger sense, these data suggest that protease maturation of HAstV, which removes 60 spikes, is not required for FcRn engagement. However, it does not rule out the possibility that protease maturation exposes another site on the HAstV virion for another host factor to bind. In fact, dipeptidyl-peptidase IV (DPP4) was previously identified as a cofactor for HAstV infection[27]. Recently, DPP4 was shown to facilitate virus attachment, whereas FcRn promotes virus internalization and genome release[46]. Interestingly, DPP4 has not been observed to bind the HAstV spike or core domain, however, it is possible that DPP4 binds to a site on the HAstV virion that the recombinant HAstV core domain does not display.

In an infection context, HAstV would encounter FcRn at the cell surface and/or in an early endosome at neutral and/or acidic pH[31]. The results of this study suggest that HAstV would bind FcRn slightly better at acidic pH compared to neutral pH. While these results can't definitively assign a specific role for FcRn in HAstV infection, increased affinity towards FcRn at lower pH could implicate FcRn in HAstV uncoating and genome release. Notably, inhibitors of Rab7, a small GTPase involved in early to late endosome maturation, block HAstV infection in Caco-2 cells, suggesting that endosomes containing HAstV mature to the late stage for HAstV to uncoat[47]. The requirement for late endosomes, along with the ability of HAstV spike to bind FcRn at pH 5 (the approximate pH of a late endosome), suggests that FcRn may be involved in uncoating HAstV. However, more work is needed to determine the mechanistic role of FcRn in HAstV uncoating.

Predictive structure programs like AlphaFold are rapidly improving. AlphaFold2 multimer was unable to confidently predict the structure of two FcRn molecules in complex with a HAstV1 spike dimer, and the model with the highest confidence score shows the HAstV1 spike dimer splitting apart to bind FcRn (Supplementary Fig. 12a). With the release of AlphaFold3, the confidence score was much higher, and the model of two FcRn molecules bound to the HAstV1 spike dimer is nearly identical to the experimentally-determined crystal structure reported here (Supplemental Fig. 12b). This highlights the major advancements in protein structure prediction that have occurred recently.

Overall, our findings illuminate a previously unrecognized site of vulnerability on the surface of the HAstV virion. Although this site is poorly conserved, FcRn mimetics and inhibitors may function as therapeutics targeting all classical HAstVs. Furthermore, to develop vaccines targeting this poorly conserved site, a multivalent approach may be required to elicit diverse antibody responses targeting the diverse neutralizing epitopes of the classical HAstV spikes.

## Methods

### Generation of HAstV capsid spike expression vectors
The expression vectors for the classical HAstV1-8 capsid spikes and the divergent HAstV-MLB1 and HAstV-VA1 capsid spikes were generated previously[22,23,25,36–38]. Briefly, codon-optimized synthetic genes encoding classical HAstV capsid spikes were cloned into the pET52b +

expression vector in-frame with a C-terminal thrombin cleavage site and a 10x histidine affinity tag. HAstV1 K467A and K514A mutant spike plasmids were generated by Phusion site-directed mutagenesis and confirmed with Sanger sequencing. HAstV1 Y475A and 3 A mutant spike plasmids were generated by GenScript. HAstV1 K467H spike mutant plasmid was cloned by GenScript into the pET52b + expression vector in-frame with a C-terminal Avitag for directed biotinylation for acidic pH binding studies. Expression plasmids can be provided with an approximate 2-month timeframe by R.M.D. at the University of California, Santa Cruz, pending scientific review and a completed material transfer agreement. Requests for these materials should be submitted to rmdubois@ucsc.edu.

### Expression and purification of HAstV capsid spikes in *E. coli*
The respective HAstV capsid spike plasmid was heat-shock transformed into T7Express *E. coli* cells (New England Biolabs). A culture was grown in LB media with ampicillin (100 µg/mL) via shaking at 37 °C. Upon reaching an $OD_{600}$ of 0.6, isopropyl β-d-1-thiogalactopyranoside (IPTG) was added to a final concentration of 1 mM and the culture was shifted to 18 °C and shaken overnight. Cells were harvested by centrifugation and resuspended into lysis buffer (20 mM Tris, pH 8.0, 300 mM NaCl, 20 mM imidazole). Cells were lysed by sonication, and lysates were centrifuged for 30 min at 40,000 g at 4 °C. The soluble fraction was filtered through a 0.2 µm Acrodisc filter (Pall) and loaded onto a gravity column with HisPur cobalt resin (Thermo Fisher Scientific). The column was washed with lysis buffer, and proteins were eluted by the addition of lysis buffer containing 500 mM imidazole. The sample was dialyzed into TBS (10 mM Tris, pH 8.0, 150 mM NaCl) overnight at 4 °C. Purity was assessed by SDS-PAGE with Coomassie Blue staining. HAstV1 spike mutants were analyzed by SEC (Superdex 200 Increase 10/300 GL) to confirm dimerization, suggesting proper protein folding.

### Expression and purification of FcRn in CHO-S cells
Codon optimized synthetic genes encoding the ectodomain of the wild-type FCGRT gene (UniProt: P55899, Met1-Ser297), the β-2-Microglobulin (β2M) gene (UniProt: P61769, Met1-Met119), or the FcRn mutants D153E, T149N_G151T, and T149N_G151T_D153E were cloned separately into a modified pCDNA3.1 vector with methylation targets in the CMV promoter deleted[48]. The FCGRT construct was cloned in-frame with a C-terminal thrombin cleavage site, TwinII-Strep tag, and AviTag. All plasmids were maxiprepped (Machery-Nagel). CHO-S cells were resuspended to a density of $2 \times 10^8$ cells/mL and electroporated with a MaxCyte STx using OC-400 cuvettes. A total of 120 µg of DNA was used with a 1:2 (w:w) ratio of FCGRT to β2M plasmid DNA. The cells were grown in CD-OptiCHO media (Gibco 12681029) with 1 mM sodium butyrate and fed daily with CHO feed (Gibco A1023401) supplemented with 7 mM L-glutamine, 5.5% glucose, and 23.4 g/L yeastolate. The cells were maintained at 32 °C, 8% $CO_2$, 85% humidity, and 135 rpm for 7 days. The cells were centrifuged, and the media was supplemented with 1 x protease inhibitor cocktail (Millipore 539137) and filtered through a 0.2 µm filter. The media was diluted 1:2 (v:v) with StrepTrap XT binding buffer (100 mM Tris pH 8.0, 150 mM NaCl, 1 mM EDTA) and loaded onto a Cytiva StrepTrap XT affinity column. The column was washed with binding buffer, and the protein was eluted with binding buffer containing 50 mM biotin. The sample was dialyzed into TBS (10 mM Tris, pH 8.0, 150 mM NaCl) overnight at 4 °C. Purity was assessed by SDS-PAGE with Coomassie Blue staining.

### Crystallization and structure elucidation of FcRn-HAstV1 spike complex
Affinity-purified FcRn and HAstV1 spike were mixed, incubated with thrombin to remove tags, and dialyzed overnight into TBS. The digestion product was loaded onto a Superdex 200 16/600 column to isolate FcRn-HAstV1 spike complex. Fractions containing FcRn-HAstV1

spike complex were pooled and concentrated to 6.65 mg/mL. The sample was crystallized via hanging-drop vapor diffusion with 0.1 M NaCl, 0.1 M Bis-Tris:HCl pH 6.1, and 1 M ammonium sulfate. The crystal was cryoprotected in a solution containing 0.2 M ammonium citrate dibasic, pH 4.8, 5% PEG 3350, and 20% glycerol, and then submerged in liquid nitrogen. Diffraction data was collected using beam line 501 at The Advanced Light Source. Two data sets from a single crystal were processed using DIALS (v.3.18). The data sets were indexed and integrated separately but symmetrized, scaled, and merged together. Molecular replacement was performed with Phaser in the PHENIX suite (v1.20.1-4487) using the previously experimentally determined structures of FcRn (PDB: 1EXU) and HAstV1 spike domain (PDB: 5EWO). Two FcRn-HAstV1 spike complexes were placed in the asymmetric unit. The structure was refined using multiple rounds of manual and global refinement in COOT (v0.9.8.92) and PHENIX (v1.20.1-4487). A paired-refinement approach with the near-final structure was used to determine the data resolution cutoff with the lowest $R_{free}$ value and finalize the model[49]. The model has been deposited in the Protein Data Bank with accession number: 9DBT. Structural figures were generated in PyMOL (v3.1.4.1).

### Biolayer interferometry binding assays

Biolayer interferometry data was collected using an Octet RED384 with the Data Acquisition Software (v.11.1.1.19). Binding assays were performed in two buffers, one at pH 7.0 (0.1 M sodium phosphate, 150 mM NaCl, and 0.05% Tween-20 at pH 7.0) and the other at pH 5.0 (0.1 M sodium acetate, 150 mM NaCl, and 0.05% Tween-20 at pH 5.0). Binding assays were performed at 25 °C with shaking at 1000 rpm. Streptavidin biosensors were used with an Avitag-biotinylated HAstV1 spike or HAstV1 spike K467H to calculate $K_D$ at pH 7.0 and 5.0 in the respective buffer. HIS1K biosensors in the pH 7.0 buffer were used for the $K_D$ calculation of the 10x histidine-tagged classical HAstV spikes, HAstV VA1 and MLB1, and the HAstV1 spike mutants, besides K467H. Biosensors were equilibrated in buffer for 600 sec. Following equilibration, HAstV spike proteins at a concentration of 3 µg/mL were loaded onto the biosensors until the binding signal reached 1.0 nm. Next, the biosensors were placed back into the buffer for 60 sec to determine a baseline before being dipped into FcRn to measure association for 120 sec. The spikes from HAstV1, HAstV2, HAstV4, HAstV5, HAstV6, HAstV8, and the HAstV1 mutant K514A were tested with a 5-point serial dilution consisting of 1000, 500, 250, 125, and 62.5 nM FcRn. At least 3 of the 5 FcRn concentrations were used for kinetics calculations. Due to lower affinity, the remaining spikes from HAstV3, HAstV7, HAstV VA1, HAstV MLB1, and the HAstV1 mutants K467A, Y475A, and 3 A were tested at a single FcRn concentration of 2000 nM, and $K_D$s were estimated from that one concentration. The spike from HAstV1 was used to test FcRn mutants. FcRn mutants D153E and T149N_G151T were tested with a 5-point serial dilution consisting of 500, 250, 125, 62.5, and 31.25 nM. Due to lower affinity, the FcRn mutant T149N_G151T_D153E was tested at two concentrations of 2000 nM and 1000 nM. After association, the biosensors were returned to the buffer for 60 sec to measure dissociation. Kinetics calculations were performed using the 1:1 binding model in the Octet Data Analysis HT software (v7). Each replicate was reference subtracted with a control containing no FcRn, aligned to the baseline, and aligned to the baseline step for inter-step correction. All binding assays were run in triplicate, and the average $K_D$ and standard deviation are reported. Graphs generated in GraphPad Prism (v10.6.1).

### Biolayer interferometry competition binding assays

Competition binding assays with IgG, HSA, and nipocalimab were performed using streptavidin biosensors in the pH 5.0 buffer. Concentrations of HAstV1 spike, IgG, HSA, and nipocalimab used were 1, 2, 5, and 1 µM, respectively. Due to differences in binding affinity, when saturating with IgG as the first association, IgG at the same concentration was added to the second association containing HAstV1 spike to prevent HAstV1 spike from outcompeting and knocking IgG off, as recommended by the Octet RED384 manufacturer Sartorius. Biosensors were equilibrated in buffer for 600 sec. Following equilibration, Avitag-biotinylated FcRn at a concentration of 1.5 µg/mL was loaded onto the biosensors until the binding signal reached 1.0 nm. Next, the biosensors were placed back into the buffer for 60 sec to determine a baseline. The biosensors were then moved into the two subsequent association steps, each lasting 300 sec to assure saturation. The control binding signal was determined by a biosensor with no protein in association 1 and the respective protein in association 2. The test biosensor's increase in signal from the end of the first association to the end of the second association was compared to the control and reported as a percentage of the control binding signal. Values ≤ 33% were considered competing, while values ≥ 66% were considered non-competing. Due to differences in binding affinity, nipocalimab was only shown as the first association, as nipocalimab in the second association would significantly outcompete the other FcRn binding proteins. Graphs generated in GraphPad Prism (v10.6.1).

Competition binding assays with antibody scFv 2D9 and 3E8 were performed using streptavidin biosensors in the pH 7.0 buffer. Biosensors were equilibrated in buffer for 600 sec. Recombinant scFvs 2D9 and 3E8 were generated as reported previously[10]. Concentrations of scFv 2D9, 3E8, and FcRn used were 60, 200, and 2000 nM, respectively. Following equilibration, Avitag-biotinylated HAstV8 spike at a concentration of 0.5 µg/mL was loaded onto the biosensors for 300 sec. Next, the biosensors were placed back into the buffer for 60 sec to determine a baseline. The biosensors were then moved into the two subsequent association steps, each lasting 300 sec to assure saturation. The control binding signal was determined by a biosensor with no protein in association 1 and the respective protein in association 2. The test biosensor signal from the second association was compared to the control and reported as a percentage of the control binding signal. Values ≤ 33% were considered competing, while values ≥ 66% were considered non-competing. The values shown are an average of triplicate assays. Graphs generated in GraphPad Prism (v10.6.1).

### Competetive ELISA assay

High-binding 96-well plates (Corning 3590) were coated with FcRn at 2 µg/mL in PBS pH 7.4 overnight at 4 °C. The plate was washed three times with PBST (PBS pH 7.4 with 0.1% Tween-20). The plate was blocked for one hour at room temperature with 5% bovine serum albumin (BSA) in PBST. The plate was washed three times with PBST. A 12-point 1:3 serial dilution of nipocalimab (MedChemExpress HY-P99037, Lot#: 708910), ranging from 125 nM to 0.71 pM, was prepared in 5% BSA in PBST and biotinylated HAstV1 spike was added to each nipocalimab dilution at a final concentration of 125 nM prior to being added to the plate. The nipocalimab/HAstV1 spike mixtures were incubated on the plate at room temperature for one hour. The plate was washed three times with PBST. HRP-Streptavidin (ThermoFisher N100) was diluted 1:12,000 in 5% BSA in PBST and incubated on the plate at room temperature for one hour to detect any bound biotinylated HAstV1 spike. The plate was washed three times with PBST. 3,3′,5,5′-Tetramethylbenzidine (TMB) (Sigma-Aldrich T0440) was added to the plate and incubated for 10 minutes in the dark prior to the addition of 1 N sulfuric acid to stop the peroxidase activity. The $OD_{450}$ was measured using a Molecular Devices SPECTRAmax PLUS plate reader using SoftMax Pro software (v4.8). Graphs generated in GraphPad Prism (v10.6.1).

### Reporting summary

Further information on research design is available in the Nature Portfolio Reporting Summary linked to this article.

## Data availability

The coordinates and structure factors for the FcRn – HAstV1 spike complex structure have been deposited in the Protein Data Bank under accession code 9DBT. The raw diffraction data has been made available at the Integrated Resource for Reproducibility in Macromolecular Crystallography 2.0 [https://www.proteindiffraction.org/project/data_9DBT/]. Previously published PDB accession codes referenced in this study; 4N0U, 9MI6, 7RK2, 7RK1, 9CN2, 9CBN. UniProt accession codes for protein sequences used in this study; P55899, P61769, Q61559, O12792, Q82446, Q9WFZ0, Q3ZN05, Q4TWH7, Q67815, Q96818, Q9IFX1. Source data are provided in this paper.

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

## Acknowledgements

This research was funded by NIH grant R01 AI144090 to R.M.D. and C.F.A., as well as NIH grant R21 AI188909 to R.M.D. Funding for the purchase of the Octet RED384 instrument was supported by the NIH S10 shared instrumentation grant 1S10OD027012-01. This research used resources of the Beamline 5.0.1 at the Advanced Light Source, a U.S. DOE Office of Science User Facility under contract no. DE-AC02-05CH11231.

## Author contributions

A.L., R.M.D. and C.F.A. designed the research. A.L., S.L., K.R.I. and L.R. performed research and data analysis. A.L., R.M.D. and C.F.A. wrote and revised the manuscript. R.M.D. and C.F.A. acquired funding.

## Competing interests

The authors declare no competing interests.
