## [Transparent Peer Review file · Nature Communications]

Structure of the human astrovirus capsid spike in complex with the neonatal Fc receptor

Corresponding Author: Dr Rebecca DuBois

Version 0:

Reviewer comments:

Reviewer #1

(Remarks to the Author)

The manuscript by Lentz et al. describes the crystal structure of FcRn in complex with HAstV1 spike and validates the structure through mutagenesis, investigates the pH dependency of this interaction, determines the binding affinity of the spike proteins of all 8 serotypes for FcRn, maps the epitopes of neutralizing antibodies to infer their neutralization mechanism, and demonstrates spike binding to both human and mouse FcRn. They further determine that spike competes with the natural FcRn ligand, IgG, and that this interaction can be blocked by a FDA-approved drug Nipocalimab. The modeling of the receptor interaction with immature or mature virions provided further important insights into HAstV biology. This is a well written manuscript with well executed experiments that are logically presented. The data are novel and provide important information for the field.

The authors comprehensively addressed prior comments. I have no further comments.

Reviewer #2

(Remarks to the Author)

I reviewed this manuscript when it was submitted elsewhere before it was transferred to Nature Communications. Based on the rebuttal letter for the previous review, which was also attached to this submission, the authors have addressed most of my concerns. However, some issues still need clarification, particularly regarding the functional relevance of the newly identified interaction surface.

In this manuscript, the authors revealed a previously unrecognized site on the surface of the HAstV virion that interacts with FcRn. They conducted several in vitro binding studies to demonstrate the importance of this newly identified site. I understand that previous studies have shown that knocking out FcRn inhibits infection by all 8 classical HAstV, indicating a strong interaction between HAstV and FcRn. However, how the newly identified site contributes to this overall inhibition remains unclear. It would be interesting to see the results of similar functional assays with mutagenesis at this binding interface.

The authors demonstrated that FcRn-spike complexes dimerize in solution. Considering there are four copies in the crystallographic asymmetric unit, is such a dimer interface also visible in the crystal? A brief discussion would be helpful.

Extended Data Table 1 lists 'Crystal 1' as the sample name. It would be better to include the full complex name and the PDB code of the deposited structure.

Reviewer #3

(Remarks to the Author)

The authors have satisfactorily addressed all my concerns, I found that the new version of this manuscript is better. I congratulate them on their work.

Pablo

Response to reviewer comments: Lentz et al.

Reviewer #1 (Remarks to the Author):

The manuscript by Lentz et al. describes the crystal structure of FcRn in complex with HAstV1 spike and validates the structure through mutagenesis, investigates the pH dependency of this interaction, determines the binding affinity of the spike proteins of all 8 serotypes for FcRn, maps the epitopes of neutralizing antibodies to infer their neutralization mechanism, and demonstrates spike binding to both human and mouse FcRn. They further determine that spike competes with the natural FcRn ligand, IgG, and that this interaction can be blocked by a FDA-approved drug Nipocalimab. The modeling of the receptor interaction with immature or mature virions provided further important insights into HAstV biology.

This is a well written manuscript with well executed experiments that are logically presented. The data are novel and provide important information for the field. The authors comprehensively addressed prior comments. I have no further comments.

We thank the reviewer again for their time in rereviewing this manuscript.

Reviewer #2 (Remarks to the Author):

I reviewed this manuscript when it was submitted elsewhere before it was transferred to Nature Communications. Based on the rebuttal letter for the previous review, which was also attached to this submission, the authors have addressed most of my concerns. However, some issues still need clarification, particularly regarding the functional relevance of the newly identified interaction surface.

We thank the reviewer again for their time in rereviewing this manuscript.

In this manuscript, the authors revealed a previously unrecognized site on the surface of the HAstV virion that interacts with FcRn. They conducted several in vitro binding studies to demonstrate the importance of this newly identified site. I understand that previous studies have shown that knocking out FcRn inhibits infection by all 8 classical HAstV, indicating a strong interaction between HAstV and FcRn. However, how the newly identified site contributes to this overall inhibition remains unclear. It would be interesting to see the results of similar functional assays with mutagenesis at this binding interface.

We agree that such experiments would be interesting and valuable but believe that they are beyond the scope of this work. We have added to the discussion section at lines 287-288 the following sentence: "Future FcRn mutagenesis studies could further support the importance of this site in HAstV infection."

The authors demonstrated that FcRn-spike complexes dimerize in solution. Considering

there are four copies in the crystallographic asymmetric unit, is such a dimer interface also visible in the crystal? A brief discussion would be helpful.

Yes, the FcRn-spike interface is visible in all four copies of the asymmetric unit. This is described at lines 110-114 and shown in Supplementary Figure 2.

Extended Data Table 1 lists 'Crystal 1' as the sample name. It would be better to include the full complex name and the PDB code of the deposited structure.

We agree and have changed the name in the table (now Supplementary Table 1) to "HAstV spike – FcRn (PDB: 9DBT)"

Reviewer #3 (Remarks to the Author):

The authors have satisfactorily addressed all my concerns, I found that the new version of this manuscript is better. I congratulate them on their work.

We thank the reviewer again for their time in rereviewing this manuscript.